# Bee colony remote monitoring based on IoT using ESP-NOW protocol

Armands Kviesis[1], Vitalijs Komasilovs[1], Niks Ozols[2] and Aleksejs Zacepins[1]

[1] Department of Computer Systems, Faculty of Information Technologies, Latvia University of Life Sciences and Technologies, Jelgava, Latvia

[2] Institute of Plant Protection Research 'Agrihorts', Latvia University of Life Sciences and Technologies, Jelgava, Latvia

## ABSTRACT

Information and communication technologies, specifically the Internet of Things (IoT), have been widely used in many agricultural practices, including beekeeping, where the adoption of advanced technologies has an increasing trend. Implementation of precision apiculture methods into beekeeping practice depends on availability and cost-effectiveness of honey bee colony monitoring systems. This study presents a developed bee colony monitoring system based on the IoT concept and using ESP8266 and ESP32 microchips. The monitoring system uses the ESP-NOW protocol for data exchange within the apiary and a GSM (Global System for Mobile communication)/GPRS (General packet radio service) external interface for packet-based communication with a remote server on the Internet. The local sensor network was constructed in a star type logical topology with one central node. The use of ESP-NOW protocol as a communication technology added an advantage of longer communication distance between measurement nodes in comparison to a previously used Wi-Fi based approach and faster data exchange. Within the study, five monitoring devices were used for real-time bee colony monitoring in Latvia. The bee colony monitoring took place from 01.06.2022 till 31.08.2022. Within this study, the distance between ESP-NOW enabled devices and power consumption of the monitoring and main nodes were evaluated as well. As a result, it was concluded that the ESP-NOW protocol is well suited for the IoT solution development for honeybee colony monitoring. It reduces the time needed to transmit data between nodes (over a large enough distance), therefore ensuring that the measurement nodes operate in an even lower power consumption mode.

Corresponding author
Aleksejs Zacepins,
alzpostbox@gmail.com

# INTRODUCTION

Monitoring, measuring and controlling systems are rapidly evolving in today's world in almost all aspects of human life. Embedded systems, the Internet of Things (IoT), remote environmental monitoring, and smart home automation are just a few of the applications that have been motivated by the creation of wireless sensor networks (WSN) (*Labib et al., 2019*). The IoT has lots of potential and offers a wide range of options to be applied in many research areas (*Nguyen, Nguyen & Ha, 2020*). The importance of IoT in research has also been emphasized by (*Doshi, Patel & Kumar Bharti, 2019*). Beekeeping, a traditional branch of agriculture, is one of the areas where IoT can be successfully applied. The combination

of beekeeping and information technologies has led to the development of precision beekeeping (precision apiculture), a sub-branch of precision agriculture, that focuses on an apiary management from the perspective of an individual bee colony, where information and communication technologies (ICT) and IoT plays a crucial role in establishing and granting the possibility to collect bee colony related data in real-time (*Zacepins et al., 2015*).

The aim of this article is to describe a developed IoT based system for honey bee colony temperature and weight monitoring, which includes five individual monitoring nodes based on ESP8266 microchips and a main node (based around ESP32 microchip) with wide-area network (WAN) interface. The ESP8266 microchips are manufactured by Espressif (https://www.espressif.com/en/products/socs/esp8266). Due to its technical specifications, performance properties, functionality, and low price, the ESP8266 is a good choice for battery operated WSNs and IoT-based systems, suggested also in a study by *Mesquita et al. (2018)*. The ESP32 (https://www.espressif.com/en/products/socs/esp32) is the successor of ESP8266 and provides more features thus the application range is much wider than that of ESP8266.

Various data transmission protocols for IoT are expected to provide acceptable performance, which can be seen from factors such as latency, power usage, transmission capacity, reliability, and security in transmitting data (*Glaroudis, Iossifides & Chatzimisios, 2020*). Connectivity features and efficient energy usage are the main factors for the future development of IoT technology and selection of the appropriate networking protocol for the specific monitoring task.

Many networking protocols are currently available for the development of the IoT systems (*Sen, Koo & Bagchi, 2018*). In general IoT communication protocols can be divided into two categories (*Al-Sarawi et al., 2017*): low power wide area network (LPWAN) protocols (such as Sigfox, LoRa, cellular network) and short range network protocols such as Wi-Fi, Bluetooth, ZigBee and ESP-NOW, to name a few.

All of the short range network protocols fall within the scope of local area communication protocols and are suitable for IoT devices that require both point-to-point and point-to-multipoint transmission. From the perspective of IoT devices, factors such as maximum range between two nodes, power usage, data exchange speed, and signal resistance to obstructions are directly related to how good a protocol is (*Eridani, Rochim & Cesara, 2021*).

Within this research, the ESP-NOW protocol was selected to enable communication between individual nodes and the main node, forming a wireless sensor network in a star topology.

Since our previous work (*Zacepins et al., 2020*) included the monitoring of bee colonies by using measurement nodes based on a Wi-Fi enabled microchip ESP8266, the main motivation was to reduce the power consumption during data transmission without the need to change the node's hardware configuration. Therefore, one of the solutions was to switch to a different communication protocol, like the ESP-NOW, that the same microchip already supports.

The ESP-NOW is a communication protocol developed by Espressif (*ESP-NOW, 2022*). In its essence, the protocol does not require Wi-Fi connectivity to exchange data between

devices, therefore does not incorporate the handshake (a relatively time-consuming) process. However, it requires device pairing to enable a secure connection in a peer-to-peer manner (*ESP-NOW, 2022*). This protocol is very fast, but limits the data packet size to be a maximum of 250 bytes (*Espressif IOT Team, 2016*).

Several studies have also investigated the maximum distance between devices connected *via* ESP-NOW. As reported by *Labib et al. (2021)*, using the ESP-NOW protocol inside various indoor structures, the maximum distance between sender and receiver is roughly 15 m, whereas the distance between ESP-NOW enabled devices in an outside environment is approximately 90 m, however in a research by *Pasic, Kuzmanov & Atanasovski (2021)* the authors stated that it is possible to have a stable communication over a longer distance, around 190 m. This was also supported by *Random Nerd Tutorials (2020)* claiming the distance to be around 220 m.

This protocol can be applied for different scenarios and in different research areas. For example, *Isnanto et al. (2020)* used ESP-NOW protocol to send agricultural soil data. The protocol was also used in a developed data logging system by *Koushik et al. (2021)* to record data about agricultural fields. In *Yukhimets, Sych & Sakhnenko (2020)*, the authors presented an approach based on ESP-NOW communication protocol capable devices for the implementation of distributed automatic control systems. Other applications include a voice communication system (*Hoang, Van & Nguyen, 2019*), wireless synchronization tasks (*Zinkevich, 2021*), aquaculture (*Den Ouden et al., 2022*) and a system for air quality monitoring, where ESP-NOW was used in additional communication mode (*Van Truong, Nayyar & Masud, 2021*). The devices using ESP-NOW are perfect candidates to form sensor networks, like the mesh network presented by *Labib et al. (2021)*.

Within this study we present the application of ESP-NOW enabled devices to monitor temperature and weight changes of the bee colonies. Bee colony weight monitoring provides one of the most important kinds of data a beekeeper can have about the colonies (*Fitzgerald et al., 2015*). Temperature measurements of bee colonies have the longest history and nowadays it seems to be the simplest and cheapest way to monitor bee colonies (*Zacepins & Karasha, 2013*). Temperature is a critical factor in colony health and is actively managed by bees applying heating or cooling to maintain a stable nest temperature (*Tautz, Heilmann & Sandeman, 2008*; *Stabentheiner, Kovac & Brodschneider, 2010*).

This research work was supported by the project HIVEOPOLIS which has received funding from the European Union's Horizon 2020 research and innovation programmes under grant agreement No. 824069.

# MATERIALS AND METHODS

## Location description

This research and measurements were carried out at LBTU's (Latvia University of Life Sciences and Technologies) apiary, located at Strazdu iela 1, Jelgava, Latvia (coordinates: 56.6630, 23.7538) starting from 01 of June 2022 till 31 of August 2022.

## Apiary description

Five honey bee (*Apis mellifera*) colonies were selected (based on a beekeeper's suggestion) for remote monitoring. Colonies were placed in different types of hives (Latvian design type and polyfoam hives). All hives were put in the same location in an open environment.

## Monitoring system
### Measurement nodes

All five colonies were equipped with a bee colony monitoring system based on the ESP8266 microchip inspired by the monitoring system developed within the SAMS (Smart Apiculture Management Services) project (*Wakjira et al., 2021*) and described in detail in *Zacepins et al. (2020)*. For weight monitoring, a single-point load cell Bosche H30A was used. The accuracy of the load cell and the A/D converter was empirically evaluated in *Kviesis et al. (2020a)*.

Two DS18B20 1-Wire® sensors were used per colony to monitor temperature. One temperature sensor (Dallas DS18B20) was installed inside the hive above the hive body (brood frames) as suggested by *Stalidzans & Berzonis (2013)* and validated by *Cook et al. (2022)*. The second temperature sensor was placed outside the hive, to monitor the ambient temperature. Each monitoring system was powered by a Li-ion 18650 3.7 V 2000 mAh battery.

Data about the bee colony and battery discharge status were collected every 30 min and sent to the main node using the ESP-NOW protocol. It should be mentioned that the ESP-NOW protocol is limited to messages up to 250 bytes. The data packet containing all measured parameters, were serialized using Protocol Buffers and was around 50 bytes in size therefore it fits well within the message size limit.

The number of individual nodes can be easily extended by simply adding new nodes. ESP-NOW protocol allows connection of up to 20 devices (including 10 encrypted peers) (*Espressif IOT Team, 2016*), which is still enough, because usually one apiary consists of 20 bee colonies. In case there are more colonies, additional main nodes could be used.

### Main node

Main node is based on the ESP32 microchip with an additional SIM800L module for the GSM (Global System for Mobile communication) SIM (Subscriber Identity Module) card, which is used for the data transfer to the remote data platform through an MQTT (a lightweight messaging protocol) broker.

The ESP32 board used in this study has a dual-core Xtensa® 32-bit LX6 microcontroller with adjustable CPU frequency (80–240 MHz) (*Espressif Systems, 2022*). In our case the operation of the main node utilizes both ESP32 cores. The ESP32 uses a modified version of the original FreeRTOS - a real-time operating system intended to be used in an embedded environment (on microcontrollers and system-on-chip (SoCs) circuits) (https://www.freertos.org/index.html). As it is described in the ESP32 documentation (*FreeRTOS, 2022*) the modified version of the FreeRTOS allows the ESP32 to use symmetric multiprocessing and therefore run multiple tasks in parallel on both cores. In our case we used this feature to assign dedicated tasks to be run on both cores in parallel. Tasks that were related to ESP-NOW message exchange with measurement nodes were assigned to Core0,

where incoming messages were added to a queue (a thread safe approach in FreeRTOS to pass messages between tasks). After a predefined time, the contents of the queue were checked and read by a task running on Core1 that also initialized the connection to a GSM network and sent the messages to the remote server using MQTT protocol. In such a way, time consuming procedures, like establishing the Internet connection and connecting to a MQTT broker, can be run in parallel without disrupting the communication with measurement nodes.

To enable the connection to the Internet, we used a standard SIM card from our local mobile operator. The main node was powered by a self-made mini solar power station containing a 12 V, 12 Ah lead–acid battery and a 12 V, 10 W solar panel together with a pulse width modulation (PWM) type solar charge controller.

### Data transfer approach

Schematic overview of the data transfer procedure is demonstrated in Fig. 1.

The five monitoring nodes are communicating with the main node _via_ the ESP-NOW protocol. After a predefined time interval (if there is any data to send) the main node connects to the Internet and transfers data using the MQTT protocol directly to the remote MQTT relay which then selects the destination data endpoint. In our case two databases were used: InfluxDB (https://www.influxdata.com/) and SAMS data warehouse (DW) (_Komasilovs et al., 2019_; _Wakjira et al., 2021_). Grafana (https://grafana.com/), an open source web application, was used to visualize data stored in InfluxDB, but SAMS user interface (UI) for data stored in SAMS DW. The screenshots below (Figs. 2 and 3) demonstrate how the summary view and a detailed view of the colony monitoring was shown to the user in real-time in both applications. In the SAMS UI's dashboard information about last measurements (weight, inside and outside temperature and battery voltage level) is shown to the user. Detailed reports are available under the "Reports" button.

In the Grafana UI (Fig. 3) all the information is presented as charts showing dynamics of the measured parameters per hive.

## Setup of distance and node connection time tests between nodes

Before the implementation in the field, we evaluated the distance between two devices using the ESP-NOW protocol. We were mostly interested to test the distance in an open environment, since the bee hives are placed in an (more or less) open field. For this purpose, we used the main node and one monitoring node. The monitoring node sent a data packet in size of 50 bytes (similarly as it would be in the bee colony monitoring case) every 5 s. The main node additionally was equipped with a display to show the received data. After the data was transmitted to the main node, the onboard LED (Light Emitting Diode) blinked and the received data were displayed on the screen, showing a successful data transfer session. Afterwards, the approximate distance was measured using the Google Maps distance measure tool.

To test the time that is required for the measurement node to establish a connection to be able to send data we compared two scenarios—using Wi-Fi protocol (like in our
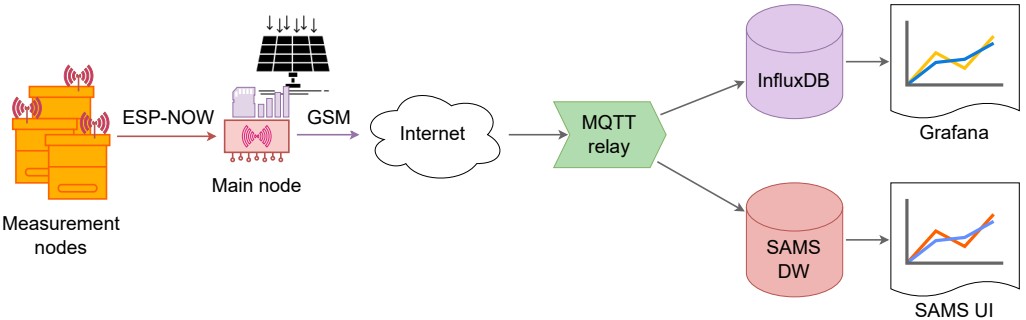

**Figure 1** **Architecture of the data transfer approach.** Schematic view of the data transfer approach, including all components, like measurement and main nodes.

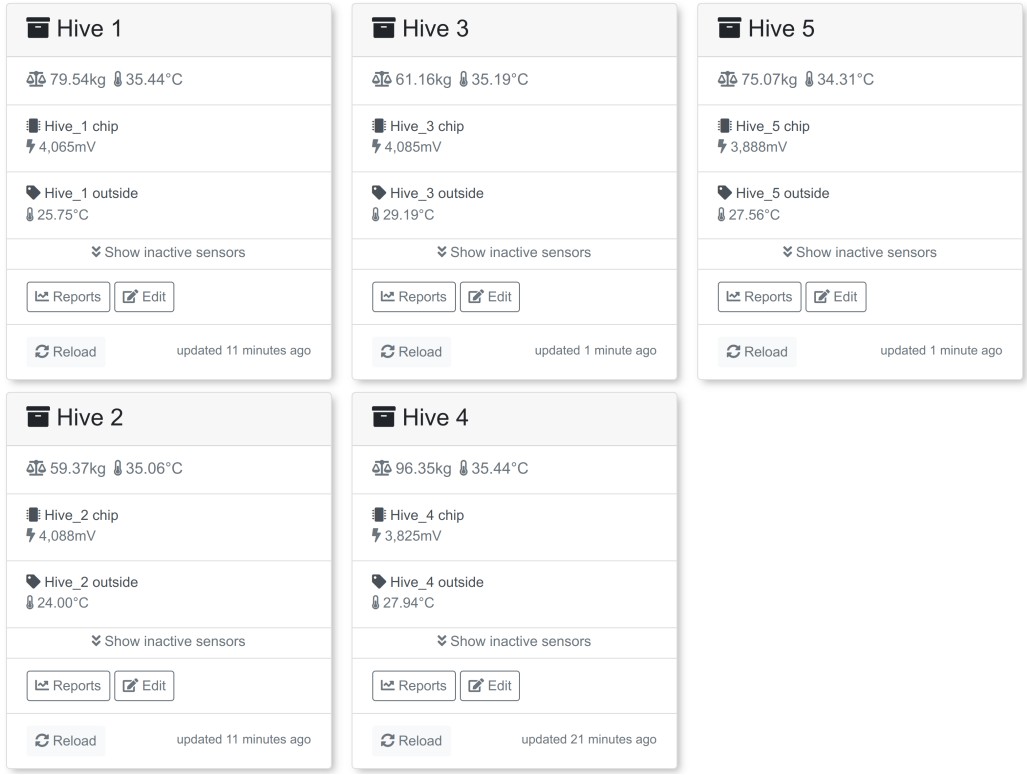

**Figure 2** **Summarized view of five colonies in the SAMS UI.** Real time data about the measured bee colony parameters presented in the SAMS UI. Data includes weight, temperature in the colony, temperature outside and battery voltage level.

previous work) and using the ESP-NOW protocol. The time was measured within the software by calling the Arduino Cores built in time function. In the Wi-Fi scenario, we were only interested in measuring the time to establish a connection with the wireless access point excluding the time to establish connection with the MQTT broker. In this way

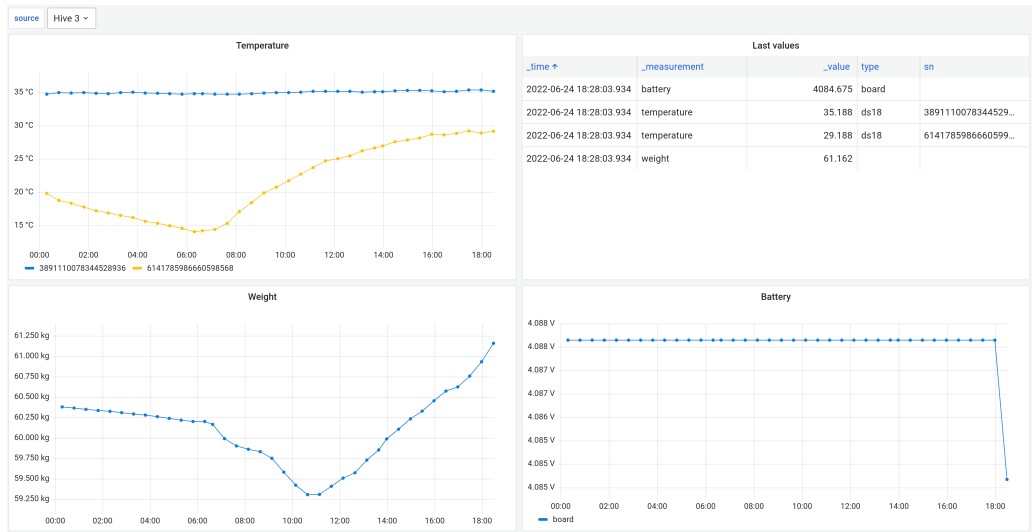

**Figure 3** **Detailed view in Grafana UI.** The bee colony data in Grafana UI. Charts include bee colony temperature, weight and node battery voltage level and last measured values.

the two scenarios could be comparable, because when using ESP-NOW, the MQTT broker is also not directly involved (the connection is managed by the main node afterwards).

# RESULTS AND DISCUSSION

## Comparison of the short range network protocols

Within this study we compared the short range network protocols using the available literature sources (*Glória, Cercas & Souto, 2017*), manufacturer data sheets and already conducted experiments by other authors (*Eridani, Rochim & Cesara, 2021*; *Random Nerd Tutorials, 2020*). The general characteristics of each protocol can be seen in Table 1.

## Distance and node connection time tests between the ESP-NOW enabled devices

Based on the performed tests, it was concluded that data can be transferred reliably up to 250 m in an open field (no obstacles) which is inline with the characteristics provided in Table 1. For the task of bee colony monitoring such a distance is more than enough, as usually bee colonies are placed closely together within one apiary.

At the experimental site, the hives with measurement nodes were placed at varying distances, where the furthest distance did not exceed 20 m. Some problems were observed with one measurement node, where the direct line of sight was being blocked by a wooden wintering building, therefore data transmission from this node was inconsistent.

Besides the distance tests we also evaluated the time needed for the measurement nodes to establish a connection with a main node in two scenarios: when using Wi-Fi (like in our previous work) where the access point was considered as the main node and, when using ESP-NOW protocol, where the main node was based on ESP32 (ref. to chapter Material and methods 3.2). On average it took approximately 5 s to wake the Wi-Fi module, set the

**Table 1 General characteristics of the short range network protocols.**

| Parameter | Bluetooth | ZigBee | Wi-Fi | ESP-NOW |
|---|---|---|---|---|
| IEEE specification | 802.15.1 | IEEE 802.15.4 | 802.11 | 802.11 |
| Frequency | 2.4 GHz | 2.4 GHz | 2.4 GHz and 5 GHz | 2.4 GHz |
| Data rate | approx. 784 Kbps | 250 Kbps | 54 Mbps | 1 Mbps |
| Maximum range | 100 m | 100 m | 100 m | 220 m |
| Maximum payload | 251 B | 128 B | 2312 B | 250 B |
| Power consumption | 1–35 mA | 1–10 mA | 100–350 mA | 60–100 mA |

wireless network credentials and establish a connection with the access point. As mentioned in chapter Material and methods 4, here we did not include the time needed to connect to the MQTT broker. In the case when using the ESP-NOW protocol, the time needed for the node to initialize the ESP-NOW, set configuration parameters and send data to the main node, was approximately 5 ms, which was significantly faster than in the first scenario.

## Power consumption
### Power consumption of the monitoring nodes

The operation of measurement nodes involves mainly deep sleep activity, when the node is consuming significantly less current compared to active mode (sensor reading and data transmission). In our case, the time it took for the node to take all sensor readings, initialize the ESP-NOW protocol and transmit data was approximately 3 s (the majority of the time the node is taking multiple weight measurements and selecting the median value), after which it was put to deep sleep (when the only active part on the ESP8266 microchip is the real time clock (RTC) module) for 30 min. During the active period, the node's current consumption was 40 mA on average, but during the deep sleep around 0.83 mA. The relatively high current consumption during deep sleep (as later found) was caused by some inefficient electronic components and the Secure Digital (SD) card module for data backup purposes. The current consumption measurements were made with the UT61D multimeter and its logging feature.

Examples of the battery discharge curves of two measurement nodes are demonstrated below (Fig. 4). The drop in both curves (on June 13 for Node 2 and on June 22 for Node 1) was due to battery replacement. As Fig. 4 shows, the monitoring nodes operated without interruptions for more than two months.

Although it should be noted that during the experiments we observed different battery discharge rates for some batteries and some measurement systems. This led to the assumptions that some of the measurement system components and batteries might be defective or behave differently outside the laboratory environment. An example of such an undesired behavior is presented in Fig. 5, where two different battery discharge rates (for the same system) are depicted. It can be seen that the system was operating for 34 days without interruptions (battery discharge rate ~15 mV/day). After the battery replacement (Fig. 5, circle 1), the system operated for about 15 days (battery discharge rate was ~17 mV/day), then the voltage started to drop dramatically, leading to a battery discharge rate of ~160 mV/day (Fig. 5, circle 2).

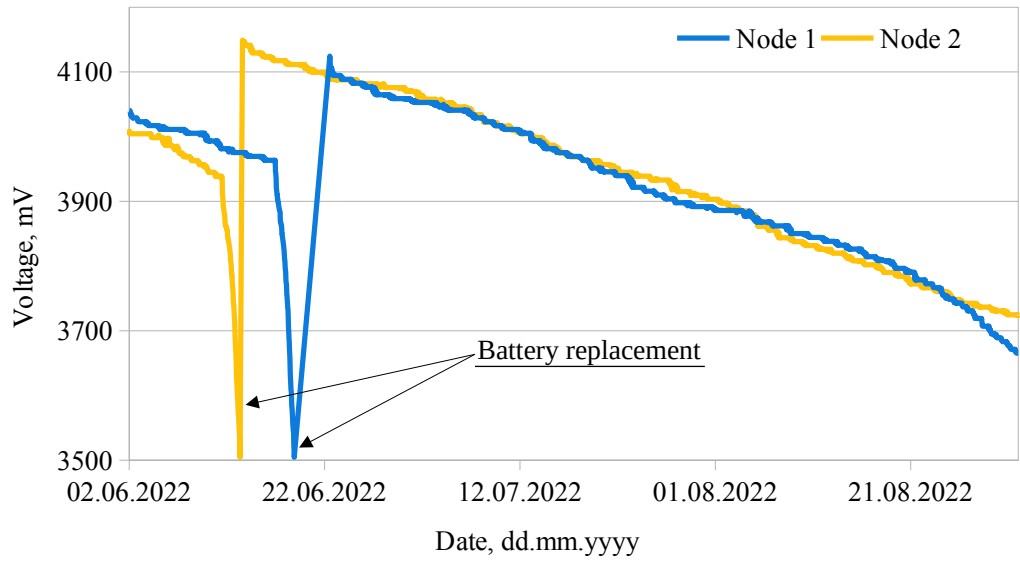

**Figure 4** **Examples of battery discharge rates.** The lines represent battery discharge rates of two measurement nodes.

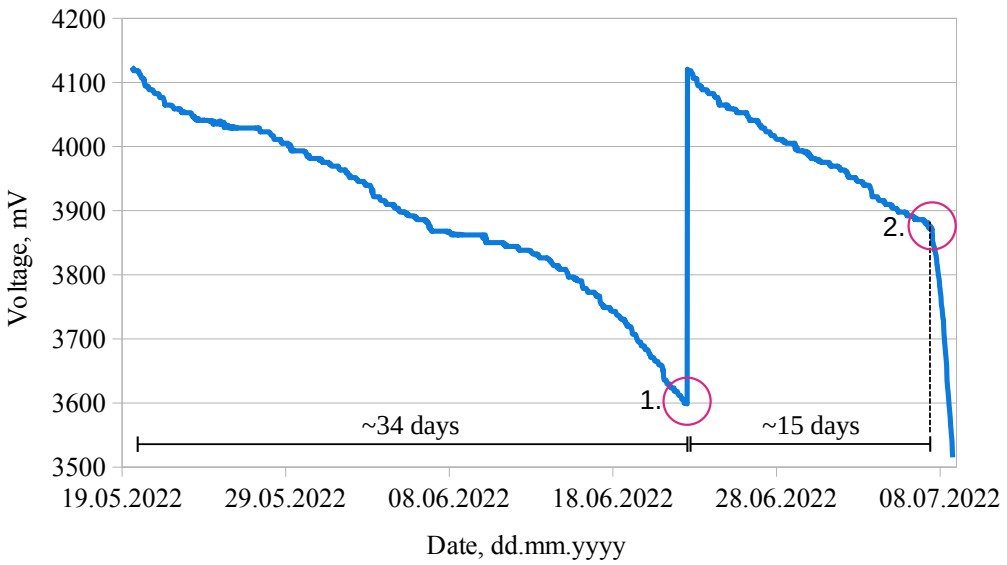

**Figure 5** **Different battery discharge rates for the same measurement node.** Different discharge rates for one measurement node suggesting defective system components or batteries.

### Power consumption of the main node

Continuous power consumption measurements of the main node during the whole experiment were not performed, but the current consumption was measured before installing the main module on field. The measured current consumption was ~100 mA. The high current can be explained by the fact that the main node is always on (the radio

frequency (RF) module is not disabled) listening for incoming messages. During the monitoring period it was observed that the main node was sufficiently powered by the attached battery and solar charging system, since there were no significant interruptions (except for some missing data points that could be due to connectivity issues) in the data, meaning it was operating all the time.

## Advantages and disadvantages of the ESP-NOW protocol

Specific communication protocols are essentially required for connecting deployed IoT devices to the network in a distributed manner. For the bee colony monitoring task ESP-NOW protocol is suitable and can be used. The ESP-NOW protocol has great distance support in LoS (Line-of-Sight) conditions with low latency. However, it consumes more power, compared to the ZigBee, and obstructions quickly weaken the signal strength. But, if a responsive device, short-range communication, and a minimal amount of data transmission is desired, ESP-NOW is suitable. For bee colony monitoring, long-range monitoring is not a critical point, as beehives are usually placed close one to another within the apiary location. With respect to the Wi-Fi protocol, it is clear that establishing a connection is quite a time and power consuming task, but it provides much higher data rates therefore is better suited for applications that require high data transfer rates, such as video streaming or large file transfers.

The fast response speed of the ESP-NOW protocol provides a lot of general advantages, like direct control of other paired devices and lower power consumption (*Amponis et al., 2022*) in a battery operated scenario.

Some disadvantages of the ESP-NOW protocol can be mentioned as well. For instance from the security aspect it is only possible to encrypt 10 nodes. The ESP-NOW is proprietary to Espressif Systems and it could be difficult and require some effort to combine it with other vendor products (like Raspberry Pi, although there have been some attempts to implement this in a Linux operating system). This can limit its interoperability with other devices and systems. ESP-NOW is designed for small-scale applications, and it may not be suitable for large-scale deployments that require a high number of devices. Another disadvantage could be the payload size that is limited to 250 bytes per message and could be a determining factor for some applications. These limitations are more general and not affecting the applicability of this protocol for bee colony monitoring purposes.

## Collected data

One aspect of this research was also focused on the urban beekeeping approach, as it was important to identify if the colony would have enough foraging resources within the city environment. This can be determined by analyzing colony weight dynamics. The collected data showed that the active foraging period started after June 22 and lasted till the first week of July, when the weight change was significant for almost every hive (see Fig. 6), except one (see Fig. 6, Hive 5), which swarmed and lost its strength. During this period the weight for colonies 1 to 5 increased by approximately 19 kg, 23 kg, 37 kg, 27 kg and 8 kg, respectively.

Additionally, the ambient temperature and temperature inside the hive was also monitored. The in-hive temperature can provide valuable information about the bee

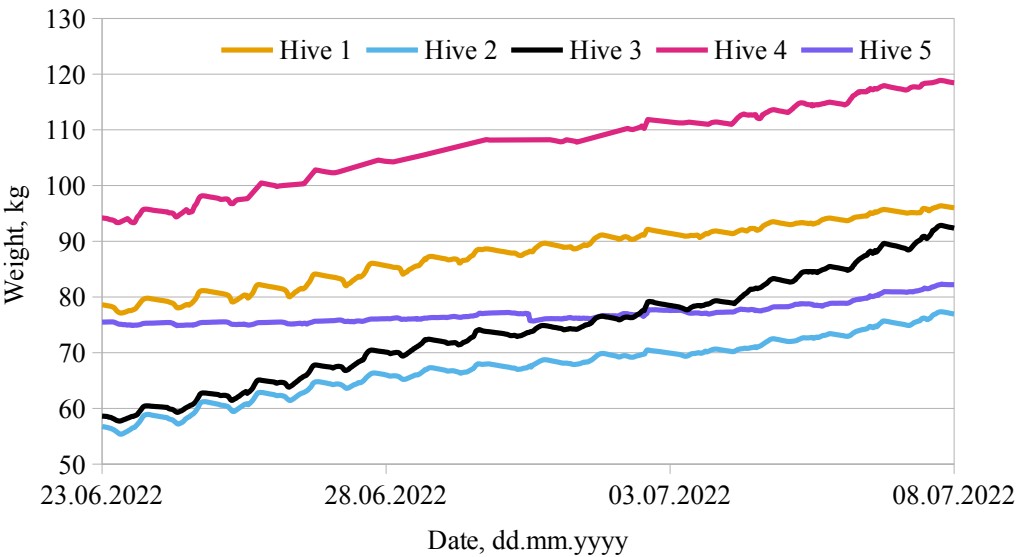

**Figure 6 Colony weight dynamics during active foraging in an urban environment.** Weight dynamics in bee colonies during active foraging period from June 22, 2022 through July 8, 2022.

colony state (*Kviesis et al., 2020b*). It should be emphasized that in order to get an insight of the colony status by temperature data (when measured by a single temperature sensor), the position of the sensor within the hive is very important. Honey bees regulate the brood temperature to be stable: from 35 to 36 °C (*Seeley, 1985*; *Bujok et al., 2002*). Previous studies (*Zacepins et al., 2011*; *Stalidzans & Berzonis, 2013*; *Kviesis et al., 2020a*; *Kviesis et al., 2020b*) have shown that it is sufficient to place a single sensor above the brood nest. An example of the in-hive temperatures during the active foraging period in our experiment is illustrated in Fig. 7. The recorded temperatures of Hive 5 were not included due to sensor misplacement after the colony swarmed. Another example of sensor misplacement can be seen in Hive 2 data, as it shows large fluctuations, although the colony was healthy (according to the beekeeper).

## Costs estimation

The economic aspect of the monitoring system is very important, so the costs should be kept as low as possible to make it financially viable for beekeepers. The system hardware components used in this experiment with approximate unit prices are summarized in Table 2. The provided costs are calculated based on the market prices in our country Latvia and can differ in other regions.

The calculated costs for the described experimental set (main node and five monitoring nodes) are 846.00 EUR. System installation, maintenance, data storage, or GSM SIM card with appropriate data plan and usage of the web system are not considered in these calculations. The total costs of the system can be reduced by using only one weight system per apiary so that the beekeeper can roughly estimate the start of the nectar flow at his apiary as well as the yield. With one weight system per apiary the cost reduction would be by 400.00 EUR.

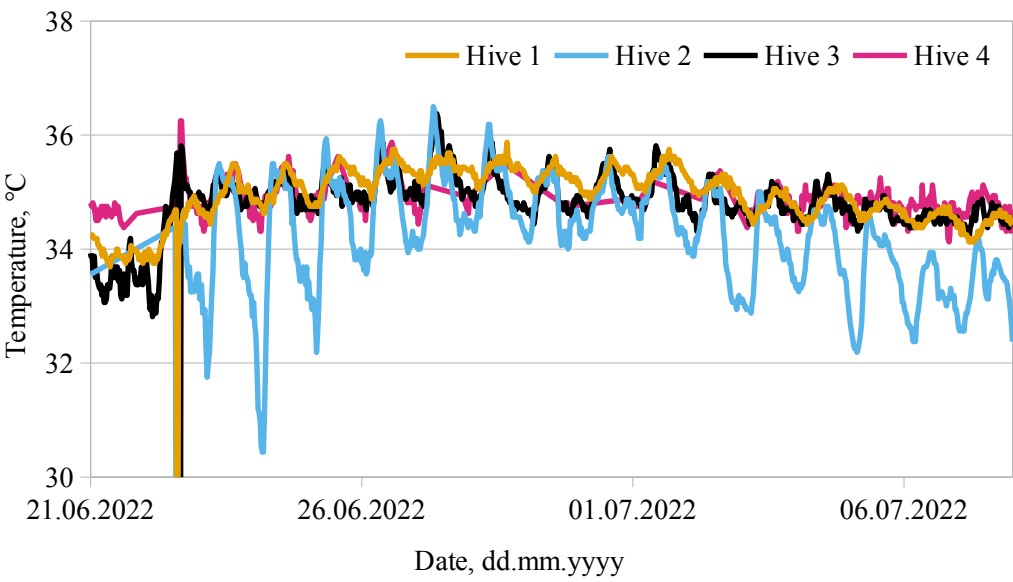

**Figure 7  Monitored in-hive temperatures.** Temperature dynamics in bee colonies for the selected period from June 21, 2022 thought July 6, 2022 are presented.

**Table 2  Cost estimation of the main and monitoring node.** Summary of costs for the development of the main and monitoring nodes.

| Nr. | Name of the component | Cost (in EUR) |
|---|---|---|
| | **Main node** | |
| 1 | ESP32 microchip based board | 25.00 |
| 2 | SIM800L module | 5.00 |
| 3 | Solar panel SOL10P (10 W, 12 V) | 35.00 |
| 4 | PWM solar charge controller | 32.00 |
| 5 | Battery GP12120 12 V, 12 Ah | 34.00 |
| 6 | Additional components | 10.00 |
| | **Main node overall costs** | **141.00** |
| | **Monitoring node** | |
| 1 | BOSCHE Wagetechnik Single point load cell H30A (200 kg) | 50.00 |
| 2 | Platform for load cell | 50.00 |
| 3 | ESP8266 microchip (ESP-12F) including adapter plate | 13.00 |
| 4 | Temperature sensor DS18B20 (x2) | 8.00 |
| 5 | A/D converter HX711 | 5.00 |
| 6 | Additional components (PCB, wires, resistors, capacitors, connectors, etc.) | 10.00 |
| 7 | Rechargeable battery Li-ion 18650 3.7 V 2000 mAh | 5.00 |
| | **Monitoring node overall costs** | **141.00** |

## System installation process

The system installation process should be as simple as possible, because usually beekeepers are not very open to new technologies and mostly not very experienced information technology (IT) specialists (*Zacepins et al., 2017*). In our case the user (beekeeper) only needs to put the scales under the hive, insert a temperature sensor into the hive and switch on the nodes, no specific configuration and assembling is required. All module configuration is done by the IT specialists before the initial installation in the field.

## CONCLUSIONS

IoT is trending in the field of ICT and builds the future of computing and communication. Implementation of the IoT principles in precision beekeeping would provide a significant impact on the beekeeping development process.

Developed honey bee colony monitoring system uses power efficient sensor nodes that consist of low cost, accurate temperature sensors, load cell for weight measurements with fast communication protocol. Such a system can be set up also in remote areas, since the main module is energy self-sufficient and communicates with the remote server over a cell network. According to literature and practical test results, the sensor nodes are able to communicate over large distances, even more than 200 m in an open field.

Developed monitoring systems can help to minimize manual bee colony inspections, which should also reduce the stress of the bee colony and increase its welfare.

Application of ICT and IoT solutions and remote monitoring systems facilitates the beekeepers' knowledge about behavior of individual bee colonies and can improve the efficiency of beekeeping management.

### Funding

Scientific research activities were conducted during the s416 project ('Evaluation and identification of the most effective control methods for topical pests of legumes and identification of factors influencing the viability of pollinators important for agriculture') which was funded by the Ministry of Agriculture of the Republic of Latvia. This research work was also supported by the project HIVEOPOLIS which has received funding from the European Union's Horizon 2020 Research and Innovation Programmes under grant agreement No. 824069. The funders had no role in study design, data collection and analysis, decision to publish, or preparation of the manuscript.

### Grant Disclosures

The following grant information was disclosed by the authors:
Ministry of Agriculture of the Republic of Latvia.
European Union's Horizon 2020 Research and Innovation Programmes:  824069.

### Competing Interests

The authors declare there are no competing interests.

## Author Contributions

- Armands Kviesis conceived and designed the experiments, performed the experiments, analyzed the data, performed the computation work, prepared figures and/or tables, and approved the final draft.
- Vitalijs Komasilovs analyzed the data, performed the computation work, prepared figures and/or tables, and approved the final draft.
- Niks Ozols conceived and designed the experiments, authored or reviewed drafts of the article, and approved the final draft.
- Aleksejs Zacepins conceived and designed the experiments, performed the experiments, analyzed the data, authored or reviewed drafts of the article, and approved the final draft.

## Data Availability

The raw data and code are available in the Supplemental Files.

## Supplemental Information

Supplemental information for this article can be found online at http://dx.doi.org/10.7717/peerj-cs.1363#supplemental-information.

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
