# Peer review of "Bee colony remote monitoring based on IoT using ESP-NOW protocol"

_PeerJ Computer Science, doi:10.7717/peerj-cs.1363_

## Round 0.1 · original submission · Major Revisions

I have received reviews of your manuscript from three scholars who are experts on the cited topic. They find the topic very interesting; however, several concerns must be addressed regarding experimental results and comparisons with current approaches. These issues require a major revision. Please refer to the reviewers’ comments listed at the end of this letter, and you will see that they are advising that you revise your manuscript. If you are prepared to undertake the work required, I would be pleased to reconsider my decision. Please submit a list of changes or a rebuttal against each point that is being raised when you submit your revised manuscript.

Thank you for considering PeerJ Computer Science for the publication of your research. We appreciate your submitting your manuscript to this journal.

Reviewer 1 ·

Basic reporting

The professional writing used throughout the paper could be improved. Specifically, numerous acronyms are used without being defined. These acronyms include, but are not limited to, FreeRTOS, ESP32, ESP-NOW, ESP8266, ICT, GSM. Not only do these acronyms need to be defined in the paper they should be redefined in the caption of any Figure or Table they appear in.

The extent to which the literature references provide sufficient background/context could be improved. Specifically, with respect to the specifications in terms of the distance of communication and speed of data exchange among state of the art protocols. The authors main contribution reported is that the use of ESP-NOW protocol as a communication technology provides longer communication distance between measurement nodes and faster data exchange. However, the capabilities of ESP-NOW in terms of each of these attributes is not reported against the leading state of the art alternatives. In addition, structuring the paper as a piece of work in the Live, Virtual and Constructive simulation community would help establish it as a computer science contribution. It currently does not have experimental depth and cannot be characterized as anything more than a case study. Reimagining it as a LVC simulation and identifying he driving concepts and relationships that are most associated with LVC systems would improve the paper. See here for more on LVC systems and how this work could be couched as such: https://journals.sagepub.com/doi/pdf/10.1177/1548512915621972)

The professional article structure, figures and tables could be improved.

With respect to article structure: the paper regularly features subsubsubsections that are preceded with a "-" (i.e. - Main node). This is very confusing to read and does not provide actual structure to the paper. Restructuring the paper to avoid such nested levels of subsections (i.e. 3 deep) would improve the readability of the paper.

With respect to figures: the figures appear to be either screenshots or visuals generated in a lossy format that cannot be resized without losing detail (i.e. jpg, gif, png). Instead, they should be rendered in a lossless visual format like pdf or eps. Also, figures 6 and 7 rely on red/green color contrast between different groups of data. There are a nontrivial number of readers who suffer from red/green color blindness. Using a color-blindness safe color palette (https://davidmathlogic.com/colorblind/#%23D81B60-%231E88E5-%23FFC107-%23004D40) and larger text fonts would improve the readability of figures. Also the authors should double check that the colors in the tables included in the paper are consistent with color-blind recommendations in the previously mentioned link.

With respect to sharing of raw data: While the raw data and source code are shared, the source code used to produce the plots is not. The paper would be improved with the inclusion of this code. In addition a README file specifying how reviewers and readers can run the provided code and a data dictionary for the provided data would improve the paper.

The paper provides a hypothesis which is, "the use of ESP-NOW protocol as a communication technology provides longer communication distance between measurement nodes and faster data exchange" however, it is not compared against any other state of the art alternatives to show that a developed bee colony monitoring system based on the IoT concept using this system can outperform any alternatives. This is a major weakness of the paper as written.

There is a lack of formal results presented with respect to the hypothesis in the paper. This should be addressed. In addition the paper uses numerous acronyms that are not defined including: ESP32, SIM800L,BOSCHE, ESP8266, PCB, etc).

Experimental design

As the paper is written it is not within the aims and scope of the journal. Specifically, the journal does not accept Case Studies / Case Reports. As written this paper is a case study / case report. The authors hypothesize that "the use of ESP-NOW protocol as a communication technology provides longer communication distance between measurement nodes and faster data exchange". However, it is not compared against any other state of the art alternatives to show that a developed bee colony monitoring system based on the IoT concept using this system can outperform any alternatives. As a result, this is not a research paper. Designing an experiment where the capabilities of ESP-NOW in terms of each of these attributes compared against the leading state of the art alternatives would address this deficiency. In addition, structuring the paper as a piece of work in the Live, Virtual and Constructive simulation community would help establish it as a computer science contribution (see here - https://journals.sagepub.com/doi/pdf/10.1177/1548512915621972)

As mentioned, while a hypothesis is specified a research question is not actually defined. It appears the authors are interested in, "To what extent does the use of ESP-NOW protocol as a communication technology provides longer communication distance between measurement nodes and faster data exchange than state of the art alternatives in a bee colony monitoring system". However, they do not state this as a research question and do not report data with respect to any alternative implementations. These are major weaknesses of the paper.

I have concerns about the ethics involved in the work the authors performed since their work was on live bees. Specifically, these concerns (a) what is the danger of exacerbating the problem of spread of parasites and pathogens to bees from academic studies using purchased colonies? (b) does the use of bee colonies give tacit approval to academia, which may be having a detrimental effect on the native populations of bumble bees and (c) is there a loss of “feeling for the organism” by researchers and particularly graduate students by treating the bees merely as pawns in an experiment. I am not the only researchers with these concerns. These along with others are enumerated in a recent paper -
Owen, R. E. (2016). Rearing Bumble Bees for Research and Profit: Practical and Ethical Considerations. Beekeeping and Bee Conservation - Advances in Research. doi: 10.5772/63048

Also, there appears to be a lot of self-citation in the paper from the authors. For example, the corresponding author is an author in 9 of the 30 (almost 1/3rd) of the non url citations in the paper. This raises serious concerns about the ethical motivations for publishing this work.

It would not be possible to replicate this work as it is performed on live insects colonies. There are simply too many degrees of variability. Furthermore, the authors do not provide a README file which specifies how reviewers and readers would run the provided source code.

Validity of the findings

The work is not compared against any alternatives. so no real conclusion about "To what extent does the use of ESP-NOW protocol as a communication technology provides longer communication distance between measurement nodes and faster data exchange than state of the art alternatives in a bee colony monitoring system" can be assessed.

While the underlying data is provided it is hard to say if it is statically sound and/or controlled. There is no comparison against alternatives in an evaluation. Furthermore, It would not be possible to replicate this work as it is performed on live insects colonies.

The chief conclusion seems to the use of ESP-NOW protocol as a communication technology provides longer communication distance between measurement nodes and faster data exchange than state of the art alternatives in a bee colony monitoring system. As I have mentioned many times the results in the paper do not support this claim. Instead the paper is a case study which is not in the scope of the journal.

Reviewer 2 ·

Basic reporting

The basic reporting of the manuscript is clear. The English language used is clear and the manuscript is quite easy to follow. The Related Work section does not provide really useful information. Mainly, Research gaps are missing in this manuscript and authors should have claimed the contribution after highlighting the research issues of existing techniques. The manuscript should give depth analysis. The structure of the paper conforms to PeerJ standards. Figures are labeled correctly and in high quality.

Experimental design

The experimental setup seems good. But the design can be evaluated with different models.

Validity of the findings

It looks that some significant findings were observed. However, the findings can be evaluated after clarifying the models.

Additional comments

In my view, the manuscript requires a minor revision before it can be published in a journal.

Reviewer 3 ·

Basic reporting

First of all, in the abstract part there should be at least a sentence about the improvement of the given method. The text in the figures must be readable. The paper can be reorganized in order to understand the impact and effectiveness of the proposed method. The paper should be carefully revised by a fluent English speaker or a professional language editing service to improve the grammar and readability.

Experimental design

There should be information about the advantages and disadvantages of the proposed technique.

Validity of the findings

There should be more comparisons with the recently proposed approaches. More analysis and discussion part are required.

---

## Round 0.2 · accepted · Accept

I am pleased to inform you that your work has now been accepted for publication in PeerJ Computer Science.

Thank you for submitting your work to this journal. On behalf of the Editors of PeerJ Computer Science, we look forward to your continued contributions to the Journal.

With kind regards,

Reviewer 1 ·

Basic reporting

The revisions made by the authors sufficiently address my concerns. The paper is now suitable for publication as the following criteria are met:

Clear and unambiguous, professional English used throughout the paper.
Literature references, sufficient field background/context are provided.
The paper presents a professional article structure, figures, tables. The raw data is shared.
The paper is self-contained with relevant results to hypotheses.
There paper uses clear definitions of all terms.

Experimental design

The revisions made by the authors sufficiently address my concerns. The paper is now suitable for publication as the following criteria are met:

The paper is a presentation of original primary research within Aims and Scope of the journal.
The research question is well defined, relevant & meaningful. It states how the research fills an identified knowledge gap.
The paper presents a rigorous investigation performed to a high technical & ethical standard.
The methods within the paper are described with sufficient detail & information to replicate.

Validity of the findings

The revisions made by the authors sufficiently address my concerns. The paper is now suitable for publication as the following criteria are met:

The paper presents its findings in a manner such that meaningful replication can be performed.
All underlying data have been provided; they are robust, statistically sound, & controlled.
The conclusions are well stated, linked to original research question & limited to supporting results.

Additional comments

The revisions made by the authors sufficiently address my concerns. The paper is now suitable for publication

Reviewer 2 ·

Basic reporting

The authors have updated the manuscript by mentioning and comparison of other protocols and discussing about the advantages and disadvantages of the ESP-NOW protocol.

Experimental design

The experimental setup seems good.

Validity of the findings

The Results and discussion section was updated, including the analysis of the advantages and disadvantages of the selected protocol.

Additional comments

The manuscript has been revised as per the reviewer's suggestions and it can be considered for publication.